# Molecular and supramolecular adaptation by coupled stimuli

Torsten Dünnebacke[1], Niklas Niemeyer[1,2], Sebastian Baumert[1], Sebastian Hochstädt[3], Lorenz Borsdorf[1], Michael Ryan Hansen [3] ✉, Johannes Neugebauer [1,2] ✉ & Gustavo Fernández [1] ✉

Adaptation transcends scale in both natural and artificial systems, but delineating the causative factors of this phenomenon requires urgent clarification. Herein, we unravel the molecular requirements for adaptation and establish a link to rationalize adaptive behavior on a self-assembled level. These concepts are established by analyzing a model compound exhibiting both light- and pH-responsive units, which enable the combined or independent application of different stimuli. On a molecular level, adaptation arises from coupled stimuli, as the final outcome of the system depends on their sequence of application. However, in a self-assembled state, a single stimulus suffices to induce adaptation as a result of collective molecular behavior and the reversibility of non-covalent interactions. Our findings go beyond state-of-the-art (multi)stimuli-responsive systems and allow us to draw up design guidelines for adaptive behavior both at the molecular and supramolecular levels, which are fundamental criteria for the realization of intelligent matter.

Biological organisms feature unique properties such as learning skills, response, and adaptation to the environment, which represent key elements determining their intelligence[1-3]. In recent years, considerable efforts have been devoted to replicating some of these features in artificial systems[4], which has culminated in the foundation of intelligent matter[5]. It has been postulated that four main functional criteria are required for the realization of intelligence in artificial matter: sensing, actuation, memory, and communication network[1]. These properties allow artificial matter to be classified in four categories of increasing complexity: structural, responsive, adaptive, and intelligent. While structural and responsive matter is state-of-the-art in the literature[4], rationalizing the origin of adaptive behavior is challenging. In fact, as evident from a recent perspective article[6], the terms responsive and adaptive are often used interchangeably in the literature due to the lack of generally applicable definitions.

In the context of materials systems, responsive behavior arises when a given system changes its properties in response to an external stimulus, whereas a regulation of properties using internal feedback is necessary for a system to be termed adaptive[5]. Apart from macroscopic systems, adaptation has been closely linked to self-organization, when referring to chemical systems interacting through reversible, non-covalent interactions[7]. Some literature sources associate adaptive behavior with the reversibility and sensitivity of non-covalent bonds to their environment and multiple stimuli[8]. The current state-of-the-art in the literature brings to light that the phenomenon of adaptation, from a chemical viewpoint, has largely focused on macroscopic systems or materials[5], i.e., films or gels. In sharp contrast, little is known about the molecular and supramolecular requirements for a given chemical entity to undergo adaptation, and how to distinguish between responsive and adaptive behavior at both the molecular and supramolecular level.

To tackle this challenge, we herein provide design guidelines towards adaptive behavior on a molecular level, and ultimately transfer this knowledge to achieve adaptation on a supramolecular level. To probe our approach, we have designed a π-conjugated molecular platform that can respond to two different stimuli (light irradiation and pH) in different ways both in the molecularly dissolved and in the assembled state. Our design comprises a central pH-responsive 2,2′-

[1]Universität Münster, Organisch-Chemisches Institut, Corrensstraße 36, 48149 Münster, Germany. [2]Universität Münster, Center for Multiscale Theory and Computation, Corrensstraße 36, 48149 Münster, Germany. [3]Universität Münster, Institut für Physikalische Chemie, Corrensstraße 28/30, 48149 Münster, Germany. ✉e-mail: mhansen@uni-muenster.de; j.neugebauer@uni-muenster.de; fernandg@uni-muenster.de

bipyridine (BPy) that is functionalized on either side with a photo-responsive cyanostilbene derivative bearing tris(dodecyloxy)benzene solubilizing groups (compound (Z)**1** in Fig. 1). The judicious choice of two distinct stimuli-responsive moieties along with molecular elements for supramolecular self-assembly has allowed us to elucidate the origin of adaptive behavior at both the molecular and supramolecular level. In a molecularly dissolved state, (Z)**1** undergoes responsive behavior towards the separate light and acid stimuli. Interestingly, when these stimuli are coupled, i.e., sequentially applied, distinct differences in the final outcome are observed depending on the order of application, i.e., 1) light + acid; or 2) acid + light, leading to molecular adaptation.

Our investigations have allowed us to classify multi-stimuli-responsive molecular entities into adaptive and non-adaptive (Fig. 2), depending on whether or not the final state of the system is affected by the sequence in which the stimuli are applied. These differences are schematically illustrated in Fig. 2, where A is the initial state of the system prior to stimuli exposure. In non-adaptive molecular systems, the application of two distinct stimuli 1 and 2 in different sequences (1 + 2 or 2 + 1) leads in both cases to the same final state C (Fig. 2, left). Therefore, both stimuli are orthogonal and their responses are independent from each other. On the contrary, if the two different sequences of stimuli (1 + 2 and 2 + 1) lead to different outcomes (state C or C'), the system exhibits molecular adaptation (Fig. 2, right). In this case, the two stimuli are coupled, and their responses are not independent but rather conditioned.

On the other hand, at supramolecular level (if (Z)**1** is allowed to self-assemble in a suitable (nonpolar) solvent), collective molecular behavior under these conditions leads to supramolecular adaptation in the presence of the individual stimuli. This conceptual work broadens the scope of single-[9–17] and dual[18–22] stimuli-responsive systems and

discloses previously elusive guidelines towards molecular and supramolecular responsiveness and adaptation, which are a prerequisite for the realization of intelligent matter.

## Results
### Preliminary considerations
In order to unravel the origin of molecular adaptation, some key aspects need to be considered: (1) the target building block shall incorporate at least two different stimuli-responsive elements, for example a light-responsive and a pH-responsive unit; (2) The application of one stimulus to a specific responsive moiety should not affect the remaining responsive unit(s); (3) there should be a coupled response of the two stimuli, i.e., the target building block should be able to process internal feedback. This is achieved if electronic communication between the responsive units takes place; (4) Additionally, for those molecular systems exhibiting collective behavior, i.e., systems that exist in a self-assembled rather than a molecularly dissolved state, the external stimulus should be able to reach the site of action even under conditions of strong aggregation.

### Initial spectroscopic investigations
The absorption spectra of (Z)**1** ($c = 1 \times 10^{-5}$ M, 298 K) in a range of organic solvents display a main transition band centered around $\lambda_{max} = 393$–402 nm resulting mainly from transitions with π-π* characteristics (Fig. 3a and Supplementary Fig. 14). In aliphatic solvents, such as $n$-octane and methylcyclohexane (MCH), a slight decrease in intensity at $\lambda_{max}$ can be observed, along with an enhancement of the absorbance at lower energies (up to 500 nm). We attribute this behavior to the onset of aggregation in these media, which appears to be amplified in $n$-octane compared to MCH, most likely due to a more efficient intercalation of solvent molecules between the solubilizing chains[23]. On the other hand, the emission spectra of (Z)**1** under

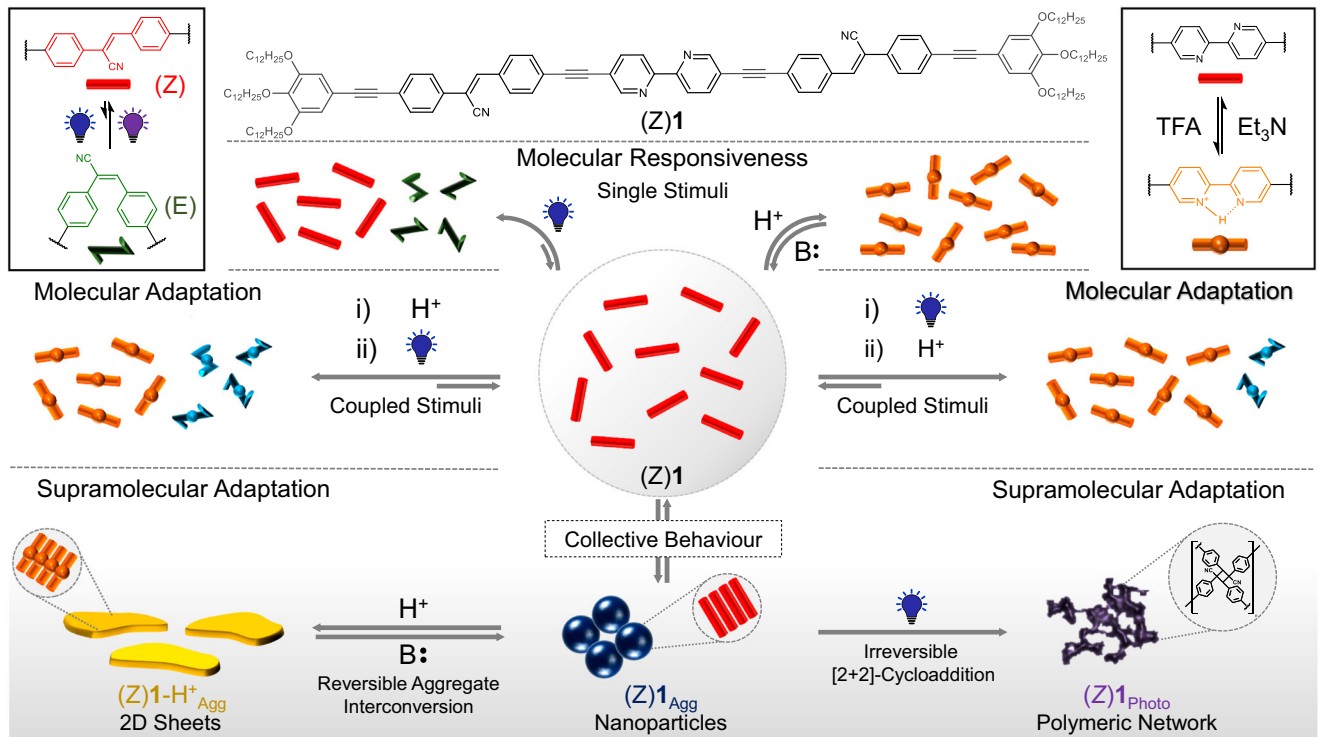

**Fig. 1 | Concepts of molecular responsiveness, molecular adaptation, and supramolecular adaptation.** Molecular design of (Z)**1** and schematic representation of its responsive behavior to independent pH and light stimuli (molecular responsiveness; top panel) or coupled stimuli (molecular adaptation; middle panel). Bottom panel: Collective molecular behavior in appropriate (non-polar, i.e.

alkanes) solvents induces supramolecular adaptation upon exposing the assemblies of (Z)**1** to acid or light stimuli. Color codes: (Z)**1**: red; (E)**1**: green; (Z)**1**-H⁺, orange; (E)**1**-H⁺, blue; H⁺: acid (TFA); B: base (Et₃N). The blue and purple light bulbs represent UV light irradiation of two different wavelengths.

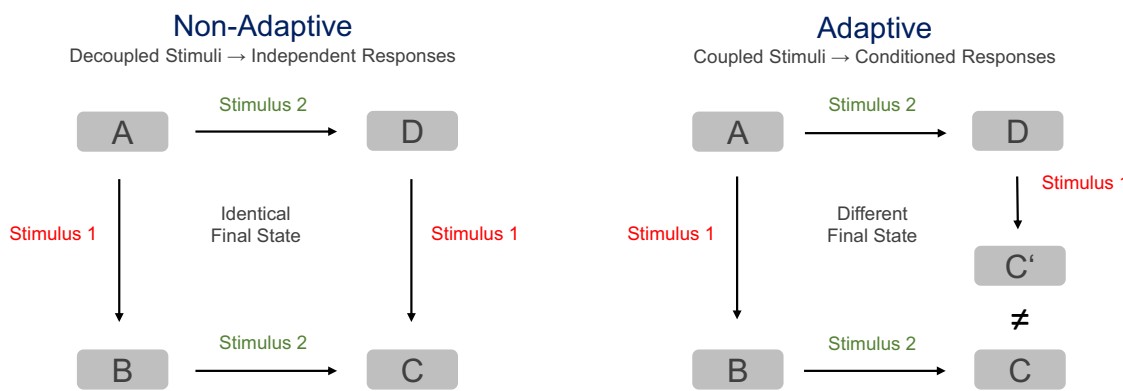

**Fig. 2 | Adaptive vs. non-adaptive behavior on a molecular level.** Conceptual representation of the classification of multi-stimuli-responsive systems into non-adaptive and adaptive proposed in this work.

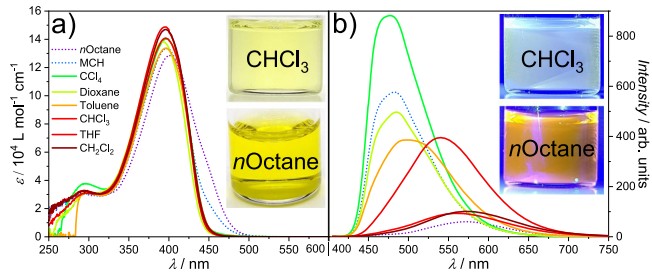

**Fig. 3 | Spectroscopic properties of (Z)1 in different solvents. a** UV-Vis spectra of (Z)**1** ($1 \times 10^{-5}$ M, 298 K) in different organic solvents in the molecularly dissolved (solid lines) and assembled (dotted lines) state. **b** Corresponding emission spectra excited at $\lambda_{max}$ of the respective absorption maxima. Insets: solutions of (Z)**1** in CHCl$_3$ and $n$-octane under daylight (**a**) and UV light (**b**).

identical conditions show remarkable differences depending on the solvent polarity (Fig. 3b, see also analysis via a Lippert-Mataga-plot[24,25], Supplementary Fig. 15). The most intense emission is observed in CCl$_4$ ($\lambda_{Em}$ (max) = 477 nm), while an increase in the solvent polarity causes a progressive quenching and a bathochromic shift of $\lambda_{Em}$ (max) (Fig. 3b and Supplementary Fig. 14b). In aliphatic solvents, such as MCH and $n$-octane, a behavior outside of the previous trend can be observed (Fig. 3b, dotted plots), which can be explained by the existence of a self-association process[26]. On the basis of these results, we identified nonpolar solvents like $n$-octane and MCH as suitable media to elicit aggregation, while the remaining solvents, such as dichloromethane (DCM) and CHCl$_3$ favor the molecularly dissolved state.

**Molecular responsiveness**

Initially, molecularly dissolved (Z)**1** was subjected to both independent, separate stimuli (acid and light). Due to the known effect of the solvent polarity on the Z/E-photoisomerization of cyanostilbenes, the photo-responsive behavior of (Z)**1** in various good solvents was tested using an LED with a core emission wavelength of 465 nm[27]. For all solvents, evidence for a Z/E-photoisomerization process can be found under initial irradiation times (Supplementary Figs. 16, 17)[28–31], as evident from the continuous hypsochromic shift of $\lambda_{max}$ and the concomitant decrease in the absorbance intensity via a defined isosbestic point (for a representative solvent (CHCl$_3$), see Fig. 4a). Analysis of the photostationary state (PSS) demonstrates that the Z/E-ratio is solvent-dependent (Supplementary Fig. 16b)[27], and the Z/E-photoisomerization is most efficient in CHCl$_3$ (Fig. 4a). According to $^1$H NMR analysis, a Z/E-ratio of 69:31 can be determined at photostationary equilibrium (Fig. 4b, for details see Supplementary Fig. 23). Although this mixture is still dominated by the Z-isomer, this state will be referred to as (E)**1** in

the following for the sake of simplicity. The photoisomerization could be further demonstrated by $^{13}$C-gated NMR studies in the PSS, in which the nitrile-carbon signal exhibits a doublet due to the typical $^3J_{C\text{-}H}$ coupling constant for E-cyanostilbenes (Supplementary Fig. 18)[27,32], while the MALDI mass spectrum shows no sign of any decomposition or side products (Supplementary Fig. 19).

Moreover, it is possible to switch back and forth between the most E-enriched ($\lambda_{LED} = 465$ nm) and the most Z-enriched ($\lambda_{LED} = 365$ nm) PSS for a minimum of ten cycles, demonstrating the reversible light-responsive behavior of (Z)**1** (Supplementary Fig. 24). Note that the irradiation time must be kept under control, particularly using LEDs of high energy, to avoid other photoreactions that may occur additionally to the Z/E-photoisomerization (Supplementary Fig. 20)[33].

Next, the effect of acid addition on molecularly dissolved (Z)**1** was tested. For these experiments, solutions of (Z)**1** in CHCl$_3$ at the same concentration as previously described for the photoirradiation experiments ($1 \times 10^{-5}$ M) were titrated against trifluoroacetic acid (TFA) ($c = 0.3$ M, 10 μL = 100 eq). Upon addition of incremental amounts of TFA, a bathochromic shift of the absorption maximum from 396 nm to 410 nm can be observed, which is accompanied by a decrease in the absorption intensity (Fig. 4c). Concomitantly, the absorbance intensity increases at lower energies (>430–500 nm), which is responsible for the intense yellow coloration of the solution upon TFA addition (inset of Fig. 4c). These spectral changes indicate the protonation of the BPy moiety of (Z)**1** by TFA[34]. The existence of two clear isosbestic points at $\lambda = 347$ and 420 nm during the TFA titration strongly suggests an equilibrium between only two species: the neutral BPy ((Z)**1**) and the mono-protonated form (BPy$^+$)[34] (in the following termed (Z)**1**-H$^+$). Monitoring the absorption at $\lambda = 450$ nm vs. the added eq of TFA reveals that the plot reaches a plateau after addition of ~1000 eq of TFA, indicating that the formation of BPy$^+$ is complete under these conditions (Fig. 4c, inset). Notably, the spectral changes associated with the protonation of the BPy can be reversed upon subsequent addition of a NEt$_3$ solution (0.3 M, 10 μL = 100 eq) while preserving the two isosbestic points (Supplementary Fig. 28b), which demonstrates the full reversibility of the protonation → deprotonation process. Time-dependent measurements after addition of NEt$_3$/TFA display no significant changes over a period of 24 h, which additionally proves the stability of (Z)**1** both under basic and acidic conditions (Supplementary Fig. 29). The spectroscopic findings are also supported by $^1$H NMR titration experiments ($4.5 \times 10^{-3}$ M, CDCl$_3$) against TFA-$d_1$ (Fig. 4d), in which the most significant chemical shifts can be detected for the BPy protons, while the signals in the aromatic region remain nearly unaffected (For a more detailed discussion, see Supplementary Fig. 25,26). We note that the initial addition of 100 eq TFA-$d_1$ causes significant signal shifts, whereas subsequent additions of TFA from 100 to 1000 eq induce only minor shifts (Fig. 4d). These results can be

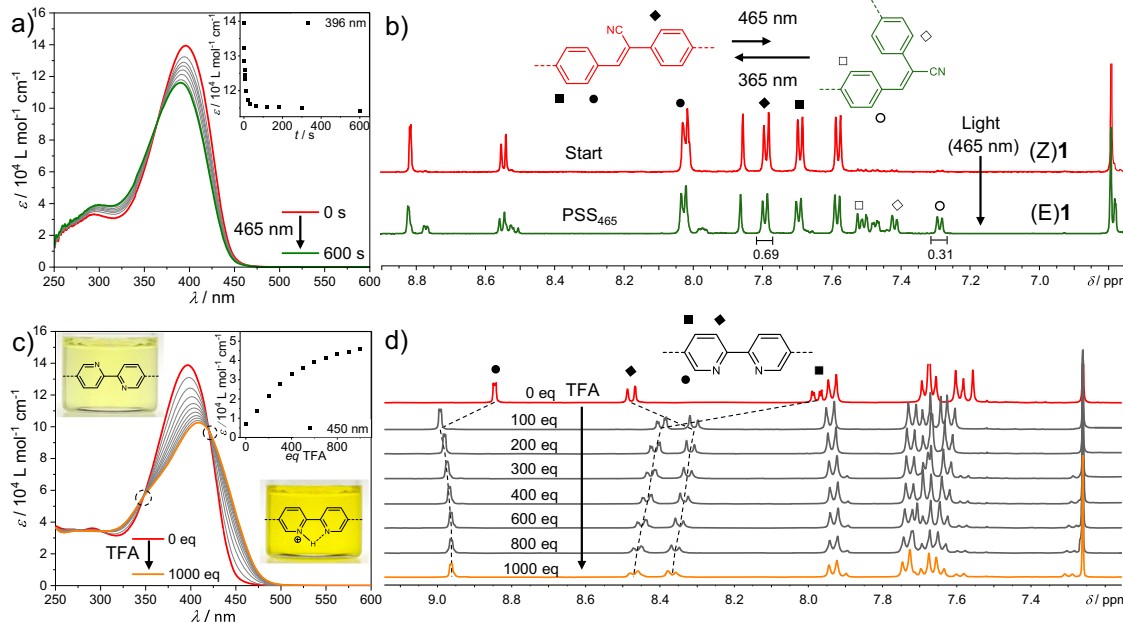

**Fig. 4 | Molecular responsiveness to independently applied stimuli. a** UV-Vis spectra of (Z)**1** ($1 \times 10^{-5}$ M, CHCl₃, 298 K) under irradiation with $\lambda_{LED} = 465$ nm. Inset: Plot of the $\varepsilon$ at $\lambda_{max}$ vs. the irradiation time. **b** Partial ¹H NMR spectra of (Z)**1** (THF-d₈, 600 MHz, 299 K). The photoirradiation studies ($\lambda_{LED} = 465$ nm) were performed in CHCl₃ ($1 \times 10^{-4}$ M, 298 K, 50 min). After reaching the PSS, the solvent CHCl₃ was removed and the obtained solid re-dissolved in THF-d₈ to achieve better separation of NMR signals. The NMR spectra of the non-irradiated and irradiated samples in THF-d₈ are then compared in (**b**). **c** UV-Vis spectra of (Z)**1** ($1 \times 10^{-5}$ M, CHCl₃, 298 K) under successive addition of TFA. Inset: plot of $\varepsilon$ at $\lambda = 450$ nm vs. the added equivalents of TFA. **d** Partial ¹H NMR spectra of (Z)**1** ($4.5 \times 10^{-3}$ M, CDCl₃, 400 MHz, 298 K) under successive addition of TFA-d₁. Color codes: (Z)**1**: red; (E)**1**: green. (Z)**1**-H⁺, orange.

explained by the increase in the overall polarity of the solvent mixture caused by addition of constant proportions of acid, in accordance with the literature[34–36]. Quantum-chemical calculations (GFN2-xTB (6.4.1)) provide insights into the most stable conformation of the BPy unit (cis or trans) of (Z)**1**-H⁺ (Supplementary Fig. 27). In accordance with the literature[37], the optimized cis-(Z)**1**-H⁺ structure is 22.6 kJ mol⁻¹ more stable than the corresponding trans-(Z)**1**-H⁺ species (Supplementary Fig. 27, Supplementary Table 1), as in the cis conformation the interaction of both nitrogen atoms with the proton provides (enthalpic) stabilization. We thus infer that a conformational change of the BPy from trans to cis occurs upon protonation[34,37,38].

## Molecular adaptation

After addressing the response of molecularly dissolved (Z)**1** to acid and light separately, we analyzed the effect of coupling these stimuli and their sequence of application. Initially, (Z)**1** in CHCl₃ was treated with an excess of TFA to produce the protonated species (Z)**1**-H⁺ (transition from red to orange spectrum in Fig. 5a), and this solution was subsequently irradiated ($\lambda_{LED} = 465$ nm) under the same conditions as described above (Supplementary Fig. 30). The resulting spectral changes (purple spectrum in Fig. 5a), even if comparatively less pronounced than for the neutral compound (Z)**1** upon photoirradiation, agree with a Z/E-Isomerization process (Fig. 5a and Supplementary Fig. 30a, b). However, the PSS of (Z)**1**-H⁺ has a higher population of the Z-Isomer compared to that of the neutral species (Z)**1** (Supplementary Fig. 31), as determined by ¹H NMR analysis (Z/E-ratio of (Z)**1**-H⁺ = 80:20, Supplementary Fig. 32). The higher Z/E-ratio for (Z)**1**-H⁺ compared to (Z)**1** can be also appreciated using various irradiation wavelengths (Supplementary Fig. 31).

We next inverted the sequence of stimuli. As previously discussed, photoirradiation of (Z)**1** ($\lambda_{LED} = 465$ nm) yields a Z/E-ratio of 69:31 in the PSS (transition from red to green spectrum in Fig. 5c). Subsequent addition of 1000 eq TFA leads to the previously described optical changes, resulting in a mixture of the protonated species (Z)**1**-H⁺ and (E)**1**-H⁺ (Fig. 5c, light blue spectrum). Notably, the final Z/E-composition obtained upon this protocol (69:31) differs from that obtained previously with reversed order of stimuli (80:20) (See Fig. 5b, Supplementary Fig. 34 and energy diagram in Fig. 5d). When this state is afterwards irradiated with the same energy ($\lambda_{LED} = 465$ nm), the system undergoes the opposite photoisomerization, ending in a more Z-enriched mixture (Supplementary Fig. 33c). Final deprotonation leads again to a mixture of the neutral (Z)**1** and (E)**1**, whose composition clearly differs from the PSS₄₆₅ of the neutral species (Supplementary Fig. 33d, 34). Thus, it could be shown that the outcome of the photoirradiation process adapts to protonation (Fig. 5). As this stimulus in the absence of light has no effect on the Z/E-ratio, the observed behavior can exclusively arise from the combination of applying both stimuli simultaneously. We note that protonation leads to a partial localization of the HOMOs and LUMOs both of the Z- and the E-form (Supplementary Fig. 35). In turn, differences in the calculated (Supplementary Tab. 2) and measured (Fig. 5a, c) absorption behavior close to the excitation wavelength become less pronounced upon protonation: In the unprotonated case, (Z)**1** shows a considerably higher absorption intensity than (E)**1**, and an absorption maximum at significantly higher wavelengths (lower excitation energies; see Supplementary Tab. 2). In the protonated case, the absorption spectra become more similar, as can be inferred from Fig. 5a, which agrees with the calculated excitation energies and oscillator strengths showing smaller differences for the protonated forms (Supplementary Tab. 2). Hence, the ratio $\varepsilon(Z)/\varepsilon(E)$ is reduced upon protonation compared to the non-protonated case. A precise quantification of this effect is difficult, as the absorption intensity at $\lambda_{LED} = 465$ nm is rather low for both of the non-protonated forms. Nevertheless, a reduced ratio $\varepsilon(Z)/\varepsilon(E)$ of the extinction coefficients upon protonation implies a higher population of the Z-form in the PSS for the protonated form compared to the non-protonated one[39], which is in perfect agreement with the experimental observations. We thus conclude that molecular adaptation arises from coupled responses.

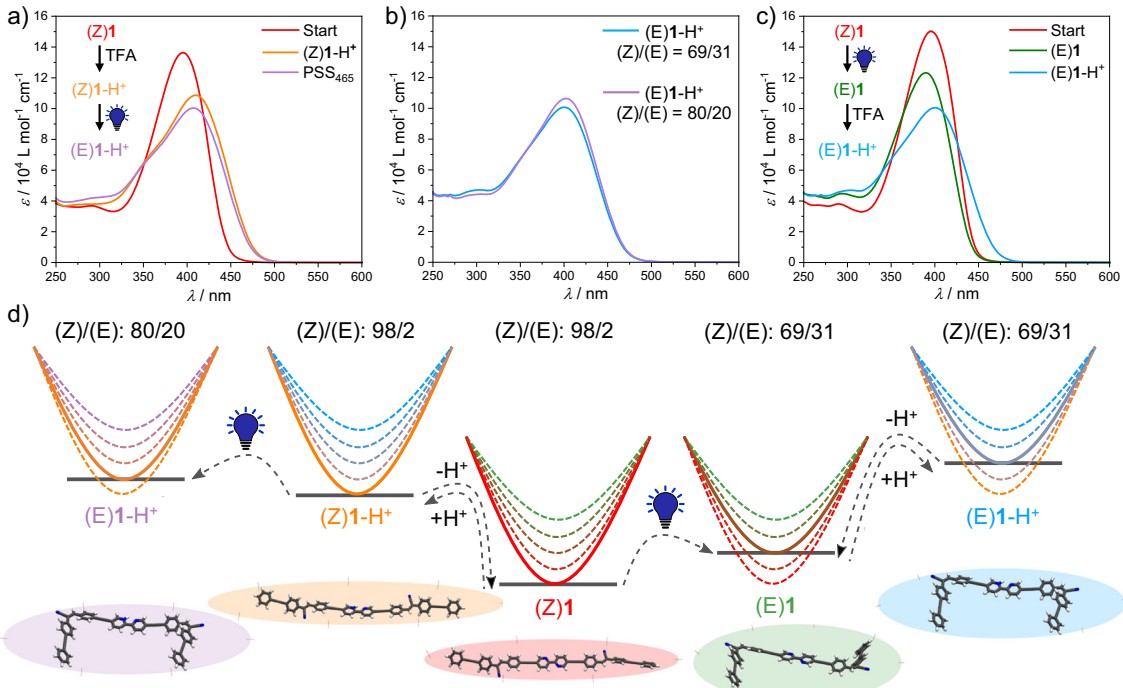

**Fig. 5 | Molecular adaptation to coupled stimuli.** UV-Vis spectra of (Z)**1** ($1 \times 10^{-5}$ M, CHCl₃, 298 K, red plot) and after application of consecutive light and acid stimuli. **a** i) acid addition (orange spectrum) + ii) photoirradiation (purple spectrum); (**c**) i) photoirradiation (green spectrum) + ii) acid addition (light blue spectrum). **b** comparison of the purple spectrum in (**a**) and the light blue spectrum in (**b**). **d** proposed energy landscape of coupled stimuli-responses on the molecular level. Color codes: (Z)**1**: red; (E)**1**: green; (Z)**1**-H⁺: orange; (E)**1**-H⁺: blue; (E)**1**-H⁺ with inverted sequence of stimuli: purple. H⁺: acid (TFA); The blue light bulb represents UV light irradiation.

## Self-assembly

Previous spectroscopic studies have shown signs of aggregation for (Z) **1** in nonpolar media. After screening multiple linear and cyclic alkanes (Supplementary Fig. 36), we selected *n*-octane as suitable solvent due to its efficient induction of aggregation, high boiling point, and colloidal stabilization of the formed assemblies. First, variable temperature (VT)-UV–Vis experiments were carried out in a concentration range between $c = 1 \times 10^{-5}$ and $1.5 \times 10^{-4}$ M using a cooling rate of 1 K/min (Fig. 6a, b, Supplementary Fig. 37). Upon decreasing the temperature from 320 K to 263 K ($1 \times 10^{-5}$ M), a slight red-shift of $\lambda_{max}$ from 392 to 395 nm accompanied by an absorption decrease can be detected (Fig. 6a). Simultaneously, the absorption increases in the area between ca. 450 nm and 500 nm, which leads to a more intense yellow color of the solution at low temperatures (inset of Fig. 6a). The minor red-shift upon aggregation suggests the formation of an assembled state (in the following denoted (Z)**1**$_{Agg}$) with a slightly slipped stacking, as observed for other cyanostilbenes[27]. This packing also agrees with the red-shift observed in VT-emission studies on cooling from 328 K to 264 K (Fig. 6c). The negligible effect of cooling rate (Supplementary Fig. 38), as well as the nearly identical cooling and heating curves (Supplementary Fig. 39), indicates that only a single assembled state is formed. The non-sigmoidal plots of the fraction of aggregated species ($\alpha_{agg}$) vs. temperature (Fig. 6b) were fitted to the nucleation-elongation model[40], yielding the following thermodynamic parameters: $\Delta H^0 = -57.20 \pm 0.26$ kJ mol⁻¹, $\Delta S^0 = -0.087 \pm 0.001$ kJ mol⁻¹ and $\Delta G^{298} = -31.33$ kJ mol⁻¹ (for further details see Supplementary Table 3).

To analyze the molecular packing of monomers of (Z)**1** within the assembly, we employed solid-state 2D ¹³C{¹H} HETCOR NMR experiments (Supplementary Fig. 40). A long cross-polarization (CP) contact time of $\tau_{CP} = 4$ ms was utilized, allowing the identification of both intra- and intermolecular ¹H-¹³C correlations (Fig. 6d, Supplementary Fig. 40b). In addition to the expected ¹H-¹³C intramolecular correlations, only (weak) additional ¹H-¹³C correlation signals between

aromatic carbons (green) and aliphatic protons (blue) occur, in agreement with the proposed stacking motif depicted in Fig. 6e. These results along with the presence of pronounced intermolecular ¹H-¹³C correlation signals between the alkoxy groups (red) rule out a large translational offset of the molecules and agree with the short-slipped stacks proposed by the UV-Vis studies. This molecular stacking results in the formation of particle-like morphologies with diameters of 120–200 nm and heights of 5–18 nm, as evidenced by atomic force microscopy (AFM) and supported by dynamic light scattering (DLS) studies (Fig. 6f and Supplementary Fig. 41,42). The relatively broad range of particle diameters observed in AFM can be correlated to convolution effects[41,42].

Quantum-chemical calculations were performed to further investigate the arrangement of the aggregated structures. For this purpose, optimized monomer, dimer, and tetramer geometries were obtained with GFN2-xTB (6.4.1)[43] omitting the solubilizing side chains. The tetramer geometry is displayed in Fig. 6e and the monomer and dimer geometries are shown in Supplementary Fig. 43. Each monomer is slightly displaced along the main aggregation axis leading to a brickwork supramolecular pattern, in agreement with the spectroscopic data. Moreover, the tetramers exhibit a tilting of the cyano-bound aromatic moieties, where the cyano groups on either side of each BPy unit are arranged in an alternating fashion, minimizing the accumulation of a larger overall dipole moment. The same holds true for the central bipyridine groups. Thus, the overall arrangement of the aromatic surfaces within the tetramer stacks in combination with their slight displacement implies that aromatic interactions are the driving force for the aggregation, which is in good agreement with the NMR and photophysical studies.

## Supramolecular adaptation

After the examination of the stimuli-responsive behavior of molecularly dissolved (Z)**1** towards light and acid as well as its aggregation

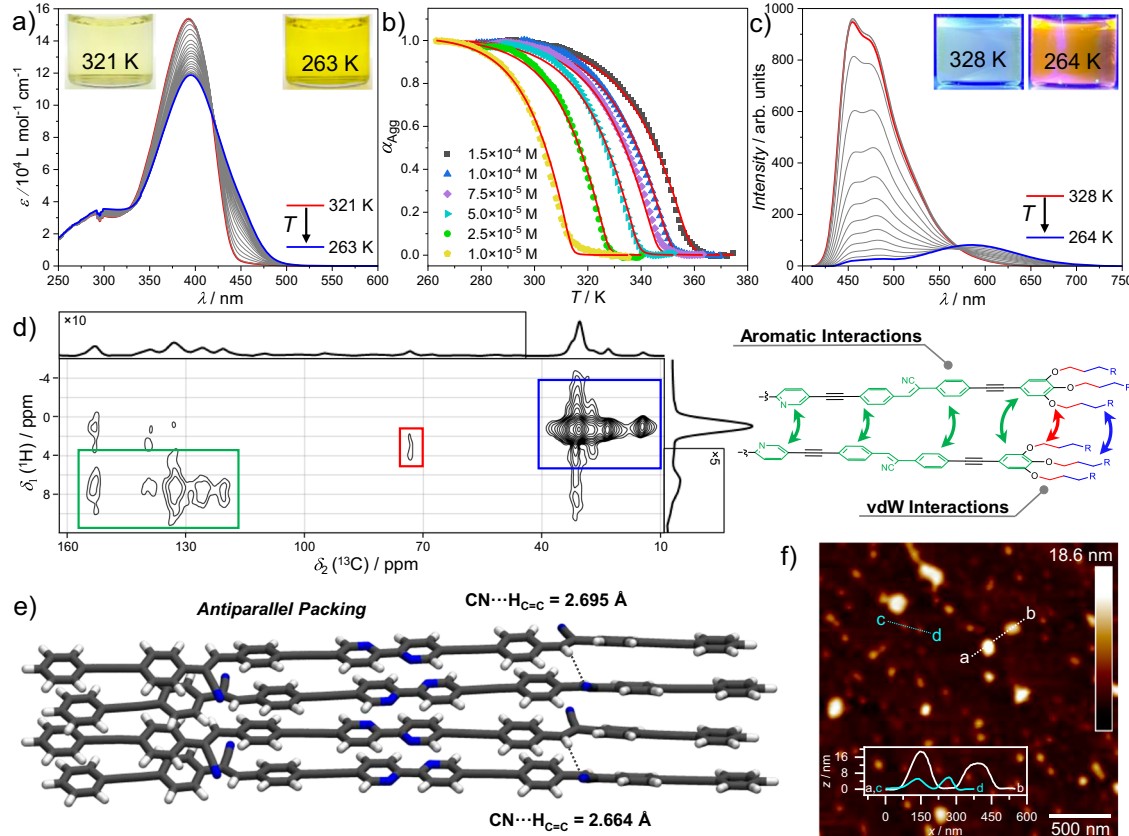

**Fig. 6 | Self-assembly behavior of (Z)1. a** VT-UV-Vis spectra of (Z)1 ($1 \times 10^{-5}$ M, 1 K/min, *n*-octane). **b** Plot of $\alpha_{Agg}$ vs. T extracted from (**a**) at different concentrations and globally fitted to the nucleation-elongation model (red plots). **c** VT-emission spectra of (Z)1 ($1 \times 10^{-5}$ M, 1 K/min, $\lambda_{Ex} = 393$ nm). **d** 2D $^{13}$C{$^{1}$H} HETCOR NMR spectrum of (Z)1$_{Agg}$ with a long CP contact time of $\tau_{CP} = 4.0$ ms, recorded at a magnetic field strength of 11.74 T. The intermolecular $^{1}$H-$^{13}$C correlations are consistent with the schematic representation of a dimer (in color). **e** Geometry optimized tetramer stack of (Z)1 (GFN2-xTB (6.4.1)). **f** AFM image of (Z)1$_{Agg}$ spin-coated onto HOPG along with height profiles of selected nanoparticle aggregates.

properties, both approaches were combined by applying a certain stimulus to the aggregated state of (Z)1 in *n*-octane ($1 \times 10^{-5}$ M). The protonation/deprotonation experiments of assembled (Z)1$_{Agg}$ were carried out analogously as described above. Upon addition of an excess of TFA, a concomitant broadening and decrease of the absorption maximum occurs, with a simultaneous bathochromic shift of $\lambda_{max} = 402$–408 nm (Fig. 7a). Concurrently, the absorption increases in the lower energy region between 500 and 550 nm, resulting in an intense orange coloration of the solution (inset in Fig. 7a). After the addition of 600 equivalents of TFA, no significant changes in the absorption spectrum can be detected (Fig. 7a, inset), indicating that the process is complete (from now this state will be termed (Z)1·H$^{+}_{Agg}$).

Subsequent stepwise addition of NEt$_3$ leads to a recovery of the characteristic UV-Vis spectrum of (Z)1$_{Agg}$ (Supplementary Fig. 44), indicating that the protonation/deprotonation process is also reversible in the aggregated state. Notably, the UV-Vis spectra obtained upon addition of an excess of TFA to the aggregated state in *n*-octane ((Z)1·H$^{+}_{Agg}$) and to the molecularly dissolved state in CHCl$_3$ ((Z)1·H$^{+}$) are remarkably different (cf. Fig. 7a and Supplementary Fig. 44), indicating the formation of an assembled state containing the protonated species at acidic conditions in *n*-octane ((Z)1·H$^{+}_{Agg}$). Based on the premise that protonation induces a trans-to-cis conformational change of the BPy units via a bridging proton and a subsequent charge accumulation, a rearrangement of the monomer units is required during the formation of (Z)1·H$^{+}_{Agg}$. The simultaneous protonation and rearrangement to form the protonated aggregate seems to be a reasonable explanation for the absence of clear isosbestic points during TFA addition. VT-UV-Vis experiments in *n*-octane ($1 \times 10^{-5}$ M, 1 K/min)

demonstrate a reversible disassembly of (Z)1·H$^{+}_{Agg}$ into monomeric (Z)1·H$^{+}$ upon heating, and a re-assembly process on subsequent cooling (Fig. 7b). Both processes show nearly identical UV-Vis spectra, pointing to the existence of a single thermodynamic species (Supplementary Fig. 45). Thermodynamic analysis of the cooling curves at different concentrations disclose weaker degrees of cooperativity ($\sigma \sim 0.1$–0.3) but dramatically higher negative values of $\Delta H^0 = -165.03 \pm 11.18$ kJ mol$^{-1}$ and $\Delta G^{298} = -52.45$ kJ mol$^{-1}$ for (Z)1·H$^{+}_{Agg}$ in comparison to (Z)1$_{Agg}$ (for further details see Supplementary Table 4). Further, the elongation temperature ($T_e$) obtained for (Z)1·H$^{+}_{Agg}$ is ca. 50 K higher than that of (Z)1$_{Agg}$ under identical conditions (Fig. 7b, inset). All these findings underline the enthalpy-driven superior thermodynamic stability of (Z)1·H$^{+}_{Agg}$ compared to (Z)1$_{Agg}$. (Z)1·H$^{+}_{Agg}$ can also be obtained by solvophobic quenching[44], upon injecting monomeric (Z)1·H$^{+}$ in CHCl$_3$ into a large volume of *n*-octane (Supplementary Fig. 46). Additionally, denaturation[45] of (Z)1·H$^{+}_{Agg}$ in *n*-octane by adding incremental volume fractions of equimolar (Z)1·H$^{+}$ in CHCl$_3$ yields the UV-Vis spectrum of the molecularly dissolved state in a single process, confirming that (Z)1·H$^{+}$ only forms one thermodynamically stable aggregate (Z)1·H$^{+}_{Agg}$ (Supplementary Fig. 47). The enhanced aggregation propensity of (Z)1·H$^{+}_{Agg}$ is also reflected in the formation of larger, two-dimensional sheets with a uniform height of 15 nm and sizes ranging from 100 to 300 nm, as imaged by AFM (Fig. 7f and Supplementary Fig. 48, 49).

Similar to (Z)1$_{Agg}$, the supramolecular organization of (Z)1·H$^{+}_{Agg}$ was examined by solid-state 2D $^{13}$C{$^{1}$H} HETCOR NMR experiments (Supplementary Fig. 50). Compared to (Z)1$_{Agg}$, no changes in terms of $^{1}$H and $^{13}$C chemical shifts occur for (Z)1·H$^{+}_{Agg}$. As for the unprotonated

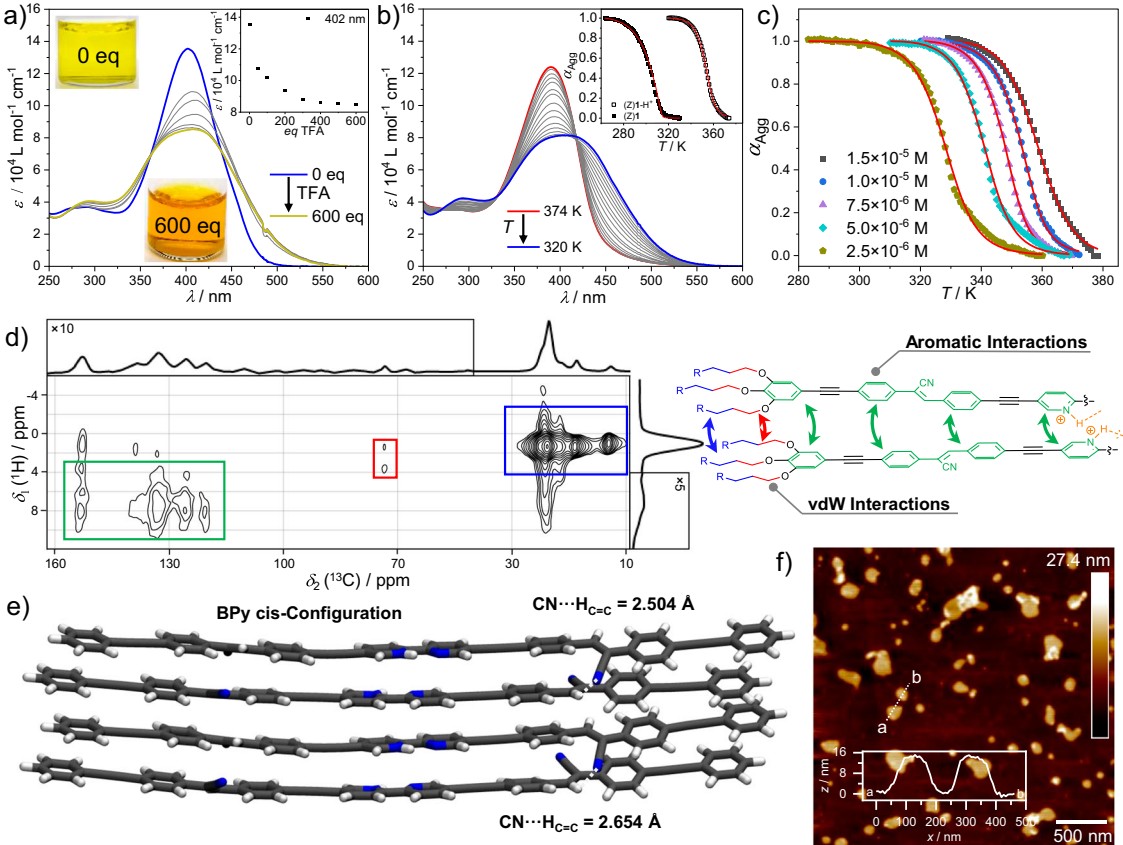

**Fig. 7 | Supramolecular adaptation to acid. a** UV-Vis spectra of (Z)$\mathbf{1}_{Agg}$ ($1 \times 10^{-5}$ M, *n*-octane, 298 K) with successive addition of TFA to generate (Z)$\mathbf{1}$-H$^+_{Agg}$. Inset: Plot of the extinction coefficient ε at $\lambda_{max}$ against the equivalents of TFA added. **b** VT-UV-Vis spectra of (Z)$\mathbf{1}$-H$^+$ ($1 \times 10^{-5}$ M, *n*-octane, 1 K/min). Inset: Comparison of $\alpha_{Agg}$ for (Z)$\mathbf{1}$ and (Z)$\mathbf{1}$-H$^+$ ($1 \times 10^{-5}$ M, *n*-octane). **c** Plots of $\alpha_{Agg}$ vs. T for (Z)$\mathbf{1}$-H$^+_{Agg}$ and fits to the nucleation-elongation model (red lines). **d** 2D $^{13}$C{$^1$H} HETCOR NMR spectrum of (Z)$\mathbf{1}$-H$^+_{Agg}$ with a long CP contact time of $\tau_{CP}$ = 4.0 ms, recorded at a magnetic field strength of 11.74 T. The intermolecular $^{13}$C-$^1$H correlations are highlighted in the colored dimer. **e** Geometry-optimized tetramer stack of (Z)$\mathbf{1}$-H$^+$ (GFN2-xTB (6.4.1)). **f** AFM image of (Z)$\mathbf{1}$-H$^+_{Agg}$ spin-coated onto HOPG along with a height profile along the line a···b.

species, only (weak) $^1$H-$^{13}$C correlation signals between aromatic (green) and aliphatic (blue) moieties can be found (Fig. 7d). Especially, the weaker $^1$H-$^{13}$C correlation signal between alkoxy groups (red) substantiates the slightly slipped stacking motif depicted in Fig. 7. Quantum-chemical calculations were performed to further investigate the arrangement within (Z)$\mathbf{1}$-H$^+_{Agg}$. Using the most favorable cis-BPy-H$^+$ conformation of the optimized monomer, dimer, and tetramer geometries were obtained using GFN2-xTB (6.4.1)[43], again omitting the solubilizing side chains. The tetramer geometry is displayed in Fig. 7e and the monomer and dimer geometries can be found in Supplementary Fig. 51. The general packing pattern is identical to the neutral compound. For the protonated tetramer, however, a bending of the aggregate backbone can be observed. In accordance with the neutral (Z)$\mathbf{1}_{Agg}$, the general ability to undergo aromatic interactions does not appear to be disturbed by the protonation. However, the vast increase in energy necessary to reach the molecularly dissolved state implies an additional driving force for aggregation. In particular, solvophobic interactions between the highly polar BPy-H$^+$ ion and the surrounding non-polar *n*-octane have a major influence on the thermodynamics of the aggregation process. Interestingly, the protonated BPy-H$^+$ cores seem to prefer an arrangement in close proximity to each other, resulting in a slight offset of the aromatic surfaces. Nevertheless, the general alignment of the aromatic systems via π-π interactions seems to remain a sufficiently strong force to control the exact positioning of the individual chromophores.

After unraveling the effect of acid stimuli on the energy landscape of (Z)$\mathbf{1}_{Agg}$, we examined the interplay between self-assembly

and photoirradiation (*n*-octane, $1 \times 10^{-4}$ M, 298 K). During the irradiation of (Z)$\mathbf{1}_{Agg}$ ($\lambda_{LED}$ = 465 nm), a strong hypochromic effect with an accompanying hypsochromic shift is observed up to an irradiation time of 690 min (Fig. 8a). This behavior, which cannot be observed under the same conditions in the molecularly dissolved state (cf. Fig. 4a), points to a photochemical transformation different from a Z/E-photoisomerization. With longer irradiation times (840 min, 14 h), a reduced overall absorbance is observed, while upon prolonged irradiation (1320 min, ~22 h) the absorption vanishes due to sample precipitation (see cuvette in Fig. 8a). Scanning electron microscopy (SEM) images of this precipitate reveal porous, network-like structures of different sizes on the silicon wafer surface (Fig. 8b and Supplementary Fig. 52). All attempts to redissolve this solid in any organic solvent failed, so did sonication or annealing at 333 K. Therefore, the resulting material obtained upon photoirradiation of (Z)$\mathbf{1}_{Agg}$ is most likely the result of a polymer network (in the following termed (Z)$\mathbf{1}_{Photo}$).

To shed some light on the newly formed material, solid-state NMR studies were performed. 1D $^{13}$C{$^1$H} CP/MAS NMR spectra for (Z)$\mathbf{1}_{Agg}$ and (Z)$\mathbf{1}_{Photo}$ (Fig. 8d, e) were recorded at 11.74 T. The $^{13}$C MAS NMR spectrum of (Z)$\mathbf{1}_{Photo}$ is characterized by a strong line broadening, in contrast to the more defined $^{13}$C signals observed for (Z)$\mathbf{1}_{Agg}$. As reported in the literature, the line widths of $^{13}$C resonances of cross-linked polymers are more pronounced than those of their corresponding straight-chain analogs and increase with the degree of cross-linking, substantiating the assumption of a network formation in this case[46,47].

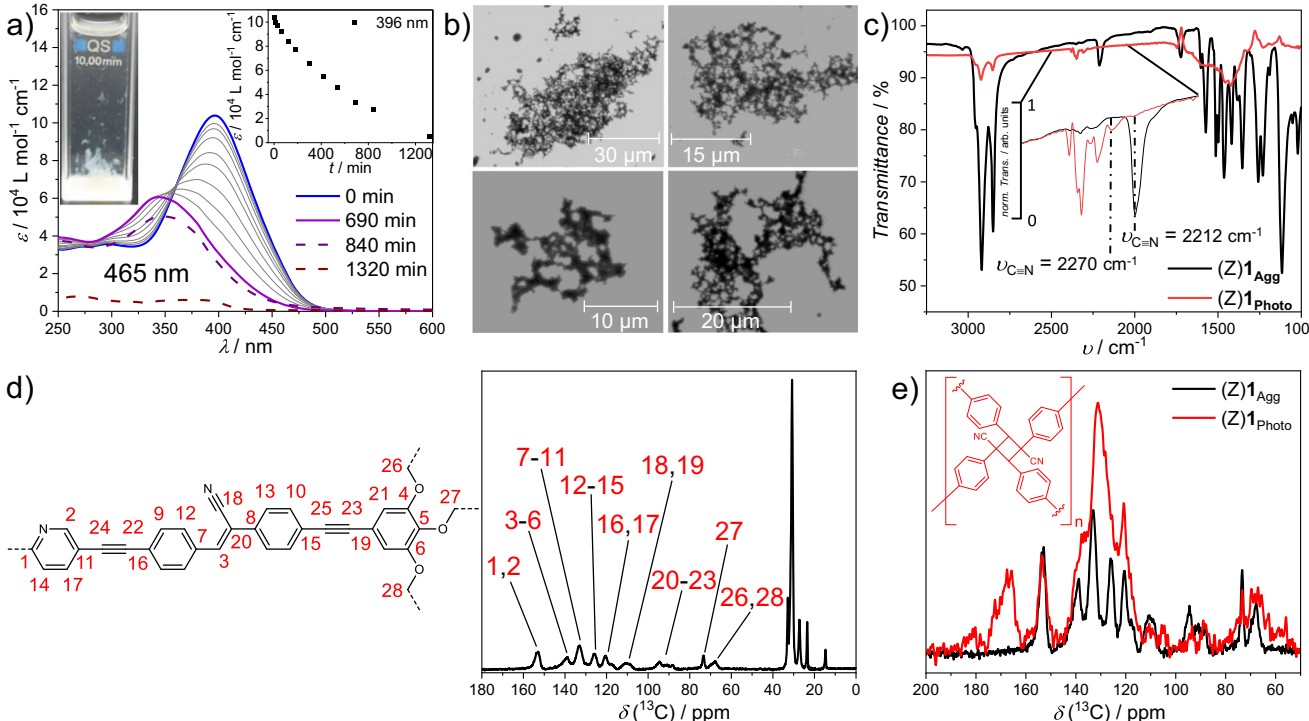

**Fig. 8 | Supramolecular adaptation to photoirradiation. a** UV-Vis-spectra of (Z)
$\mathbf{1}_{Agg}$ ($1 \times 10^{-5}$ M, $n$-octane, 298 K) under irradiation with $\lambda_{LED} = 465$ nm. Insets: Photograph of the cuvette and evolution of ε at $\lambda = 396$ nm against the irradiation time. **b** SEM images of the yellowish precipitate (Z)$\mathbf{1}_{Photo}$ after irradiation. **c** FTIR spectra

(ATR, solid-state) of (Z)$\mathbf{1}_{Agg}$ (black) and (Z)$\mathbf{1}_{Photo}$ (red). **d** $^{13}$C{$^1$H} CP/MAS NMR spectrum of the solid-state aggregate of (Z)$\mathbf{1}_{Agg}$ with corresponding assignments of the carbon signals. **e** Aliphatic region of the 1D $^{13}$C{$^1$H} CP/MAS NMR spectra for (Z)$\mathbf{1}_{Agg}$ (black) and (Z)$\mathbf{1}_{Photo}$ (red).

FTIR measurements of (Z)$\mathbf{1}_{Photo}$ show a more pronounced line broadening and reduced transmittance than that of (Z)$\mathbf{1}_{Agg}$ (Fig. 8c). In particular, the region of the C≡N stretching band relevant for the possible [2 + 2] cycloaddition is of interest. In (Z)$\mathbf{1}_{Agg}$, the band at $v_{C≡N} = 2212$ cm$^{-1}$ can be confidently assigned to the C≡N stretching of the cyanostilbene unit, in good agreement with the literature[27,48,49]. For the photoproduct, this band is shifted to $v_{C≡N} = 2270$ cm$^{-1}$ with a strongly diminished intensity. A shift in the $v_{C≡N}$ stretching to higher energies is typically observed for nitriles changing from conjugated to aliphatic systems[50,51]. The simultaneous decrease in transmission intensity can also be attributed to the lack of conjugation and the absence of electron-donating groups[52]. From the overall data collected, (Z)$\mathbf{1}_{Photo}$ has most likely formed by cross-linking of the cyanostilbene units of (Z)$\mathbf{1}$ via [2 + 2] photocycloaddition reactions. From the calculated geometry-optimized structures of (Z)$\mathbf{1}_{Agg}$, it becomes evident that the vinyl C=C double bonds are located in spatial proximity, but do not adopt an ideal pre-organization for the [2 + 2] cycloaddition. As a result, several diastereomers of the cyclobutane derivatives are most likely obtained upon photo-cross-linking, resulting in an insoluble material. Thus, once (Z)$\mathbf{1}_{Agg}$ has been exposed to light, the system remains inert and cannot respond to subsequent stimuli, such as TFA. Therefore, unlike the combination of self-assembly and acid addition, the approach presented here is an example of irreversible supramolecular adaptation. Finally, the sequence of stimuli applied to the aggregated state was inverted, i.e., (Z)$\mathbf{1}_{Agg}$ ($1 \times 10^{-5}$ M, $n$-octane) was first treated with 1000 eq TFA to obtain (Z)$\mathbf{1}$-H$^+_{Agg}$ and this species subsequently irradiated with light ($\lambda_{LED} = 465$ nm, Supplementary Fig. 53). Although the system seems to undergo a controlled photoreaction upon short irradiation times (up to 600 s) (Supplementary Fig. 53a), a strong drop in absorption occurs above this threshold (Supplementary Fig. 53b), suggesting a decomposition of the molecules under these conditions. These results

underline the importance of the stimuli sequences also on a supramolecular level.

## Discussion

In aggregation-inducing solvents (e.g., $n$-octane), (Z)$\mathbf{1}$ undergoes self-assembly into a thermodynamically more stable state (Z)$\mathbf{1}_{Agg}$ (Fig. 9). However, interfacing self-assembly and a stimulus (light or acid) allows modification of the energy landscape, thereby enabling adaptive behavior: upon addition of TFA, (Z)$\mathbf{1}_{Agg}$ undergoes rearrangement into a thermodynamically more stable aggregate species (Z)$\mathbf{1}$-H$^+_{Agg}$ via protonation of the BPy groups (Fig. 9). As (Z)$\mathbf{1}$-H$^+_{Agg}$ can be transformed back to (Z)$\mathbf{1}_{Agg}$ upon addition of a base (Et$_3$N), this system represents an example of reversible supramolecular adaptation. Similarly, (Z)$\mathbf{1}_{Agg}$ can also adapt its structure to light changes. However, in contrast to acid stimuli, the final state obtained upon photo-irradiation (Z)$\mathbf{1}_{Photo}$ is a static covalent polymer, which does not further react to external stimuli or environmental changes. As a result, (Z)$\mathbf{1}_{Agg}$ undergoes irreversible supramolecular adaptation to light.

On the other hand, collective molecular behavior is absent in solvents favoring the molecularly dissolved state, such as CHCl$_3$, due to efficient solvation. In these media, (Z)$\mathbf{1}$ exhibits conventional responsive behavior towards the two separate stimuli pH and light: upon photoirradiation with $\lambda_{LED} = 465$ nm and subsequently with $\lambda_{LED} = 365$ nm, (Z)$\mathbf{1}$ experiences a reversible Z/E-photoisomerization for multiple cycles. Alternatively, protonation of the bipyridine of (Z)$\mathbf{1}$ to create (Z)$\mathbf{1}$-H$^+$ can be achieved by addition of an excess of TFA, and subsequent addition of base (Et$_3$N) reverts the process to the neutral form. Therefore, the system exhibits reversible molecular responsiveness to independent pH and light stimuli in good solvents. Strikingly, if these stimuli are coupled, i.e., sequentially applied, the final outcome of the system depends on their sequence of application: if photoirradiation precedes the addition of acid, a final Z/E-ratio of 69:31 is achieved, whereas this isomer composition changes to 80:20 if the

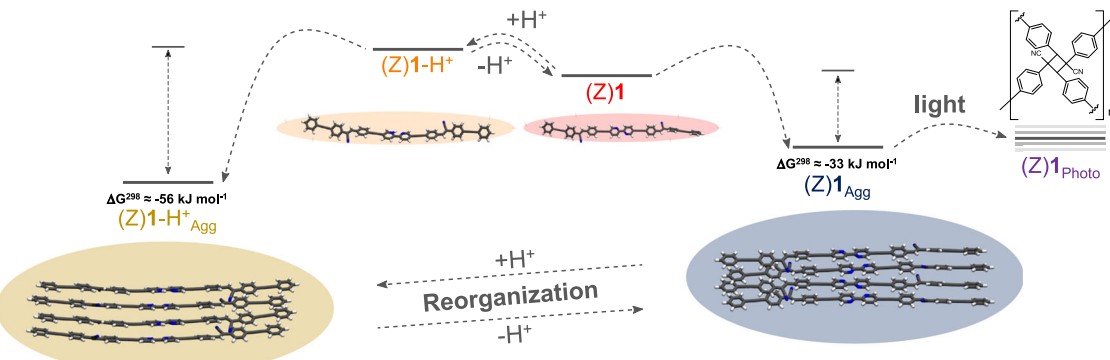

**Fig. 9 | Supramolecular adaptation to independently applied stimuli.** Proposed energy landscape of the investigated system in the aggregated state upon exposure to light (465 nm) and TFA. Color codes: (Z)**1**: red; (Z)**1**-H⁺: orange; (Z)**1**-H⁺_Agg: light brown; (Z)**1**_Agg: grey.

order of stimuli (protonation followed by photoirradiation) is reversed. Notably, when molecularly dissolved (Z)**1** is subjected to a full cycle of photoirradiation → protonation → photoirradiation → deprotonation, the final state of the system differs from the untreated sample. Thus, (Z)**1** undergoes molecular adaptation as a result of coupled stimuli: should coupling between the stimuli not occur, the sequence of application would not affect the final outcome (Z/E-ratio). Also, a full cycle of four subsequently applied stimuli (Z)**1** → (E)**1** → (E)**1**-H⁺ → (Z)**1**-H⁺ → (Z)**1** would lead to the same initial state (untreated (Z)**1**), which is however not the case.

On the basis of the gathered knowledge, we can conclude: (1) at the molecular level, responsive behavior is obtained if only one single stimulus is applied at a time; (2) coupling different stimuli may induce molecular adaptation by modifying the final outcome of the system, be it the energy landscape or the population of states; experimental evidence suggests that electronic communication between the stimuli-responsive units is required in order to achieve coupled stimuli-responses; (3) at least one of the stimuli-responses should be reversible in order to condition the remaining response(s); (4) in the assembled state, collective molecular behavior may cause distinct minima in the energy landscape, i.e., supramolecular adaptation, when a stimulus is applied. Thus, unlike the molecularly dissolved state, a single stimulus suffices to create adaptive behavior. This can be explained by the inherent collective properties of an assembled state that are intimately dependent on internal (concentration changes) or external parameters (a suitable solvent, temperature, or mechanical changes) to be formed. Thus, non-covalent self-assembly provides the required reversibility to achieve conditioned responses. In other words, the reversibility of non-covalent interactions at the supramolecular level plays the same role as the reversible stimuli-responsive unit at the molecular level; (5) the application of coupled stimuli at the supramolecular level provides access to multiple (kinetically controlled) states, thereby enriching the energy landscape of the system. In this regard, an interesting approach for further studies would be the addition of aggregation-inducing solvents (in this case *n*-octane) to a building block in a good solvent (in this case CHCl₃) to which one of the stimuli has been already applied, for instance light (Supplementary Fig. 54). This approach, which is currently underway in our group for a range of multi-stimuli-responsive monomer units, would allow for multiple aggregation states depending on the Z/E-isomer composition, opening the door to highly sophisticated adaptive supramolecular systems. In addition, we are currently optimizing the molecular design to improve the photoisomerization efficiency and gain further understanding about the concept of molecular adaptation, for example by reducing the size of the π-surface and restricting ourselves to a single photochromic unit.

Our results shed light on the nontrivial concepts of coupled stimuli, conditioned responses, and adaptation in chemical systems both from a molecular and supramolecular standpoint, which are key elements of intelligent matter. Understanding these phenomena is thus a fundamental requirement for the development of advanced smart materials.

## Methods

### NMR measurements
¹H- and ¹³C-NMR spectra were recorded either on a Bruker Avance II 400 (¹H: 400 MHz; ¹³C: 100.6 MHz) or a Bruker NEO 400 (¹H: 400 MHz; ¹³C: 100.6 MHz). Additional 1D-¹H-, ¹³C-, as well as 2D-H, H-COSY, C, H-gHSQC-, C, H-gHMBC-, F, H-HOESY- and H, H-ROESY spectra were recorded on an Agilent DD2 500 (¹H: 500 MHz, ¹³C: 125 MHz) and an Agilent DD2 600 (¹H: 600 MHz, ¹³C: 150 MHz) at a standard temperature of 298 K in deuterated solvents.

### Solid-state NMR measurements
Solid-state NMR experiments were carried out on a Bruker Avance NEO ($v_L$(¹H) = 500.39 MHz, 11.74 T) spectrometer with a 4 mm H/F/X MAS DVT probe. Samples were packed into 4 mm ZrO₂ rotors, sealed with a Vespel© top and bottom caps.

### Mass spectroscopy
MALDI-mass spectra were recorded on an Autoflex Speed manufactured by Bruker Daltonics. A SmartBeamTM NdYAF-Laser with a wavelength of 335 nm was used. ESI accurate masses were measured on a MicroTof (Bruker Daltonics, Bremen) with loop injection. ESI mass spectra were recorded on an LTQ Orbitap LTQ XL (Thermo-Fisher Scientific, Bremen) with nanospray (alternatively HPLC, loop injection, syringe pump).

### UV-Vis spectroscopy
All UV-Vis spectra were recorded on a V-770, V-750, and a V-730 by the company JASCO or a Cary 4000 by the company Agilent with a spectral bandwidth of 1.0 nm and a scan rate of either 400 or 1000 nm min⁻¹. Glass cuvettes with a path length of 1 cm, 0.5 cm, 1 mm and 0.1 mm were used. All measurements have been conducted in solvents from commercial sources from spectroscopic grades.

### Fluorescence spectroscopy
The fluorescence spectra were recorded on a FP-8500 or a FP-8550 by the company JASCO using quartz cuvettes (SUPRASIL®, Hellma) of 1 cm thickness and lamp intensity of the 75 W xenon lamp (type Ushio UXL-75 XE).

### FT-IR spectroscopy
All FT-IR spectra were recorded on a JASCO-FT-IR-4600. Solid state FT-IR was measured using a JASCO ATR Pro-One single reflector ATR unit.

## Dynamic light scattering

All DLS spectra have been recorded on a CGS-3 Compact Goniometer System manufactured by ALV GmbH, equipped with a HeNe Laser with a wavelength of 632.8 nm (22 mW) and an ALV/LSE-5004 Digital Correlator by ALV GmbH.

## Atomic force microscopy

The AFM images have been recorded on a Multimode®8 SPM System manufactured by Bruker AXS. The used cantilevers were AC200TS by Oxford Instruments with an average spring constant of $40\ N\ m^{-1}$, an average frequency of 320 kHz, an average length of 117 μm, an average width of 33 μm and an average tip radius of 8 nm in soft tapping mode. All samples were immobilized either on an HOPG or Si-wafer surface. The samples were applied in solution by placing 10 μL of a $1 \times 10^{-5}$ M solution on the substrate for 12 s followed by spin-coating at a spin velocity of 2000 rpm.

## Irradiation methods

Irradiation-based experiments were performed with LEDs by *Conrad Electronics* as follows: HighPower-LED Grün 87 lm 130° 3.8 V 1000 mA Roschwege LSC-G ($\lambda_{LED} = 520$ nm), HighPower-LED Blau 31 lm 130° 2.3 V 700 mA Roschwege LSC-B ($\lambda_{LED} = 465$ nm), HighPower-LED 365 nm 100° 4.1 V 700 mA Roschwege ($\lambda_{LED} = 365$ nm) and HighPower-LED 405 nm 100° 3.8 V 1000 mA Roschwege ($\lambda_{LED} = 405$ nm).

## Scanning electron microscopy

The SEM images have been recorded on a *Thermo Fisher Scientific Phenom ProX* Desktop SEM by Thermo Fisher Scientific. The individual images have been recorded using a zoom between 22500× and 300× with either a backscattered-electron detector (BSD) or SED detector and an acceleration voltage of either 5 or 10 kV. A BSD or a secondary electron detector was used. The corresponding samples were prepared by drop-casting a small volume (10 μL) of the sample solution on a Si-wafer surface followed by slow solvent evaporation under ambient conditions.

## Theoretical calculations

All optimized geometries were obtained with GFN2-xTB 6.4.1[43]. Electronic energies $E_{tot}$ for the protonated monomers were calculated using TURBOMOLE 7.5.1[53] and PW6B95[54] as an approximation for the exchange-correlation functional, a def2-TZVP[55] basis set and D3 dispersion correction[56] with Becke–Johnson damping[57]. Solvation-free energies ($G_{solv}$, 298 K, DCM) were obtained with COSMO-RS[58,59] using TURBOMOLE 7.5.1 for the SCF calculation and the BP86[60,61]/def2-TZVP[55] parametrization. Thermostatistical corrections to the free energy ($G_{therm}$, 298 K) were calculated from vibrational frequencies obtained with GFN2-xTB 6.4.1. Final free energies $G$ are obtained as a sum of $E_{tot}$, $G_{solv}$, and $G_{therm}$. Excitation energies and the corresponding orbital analysis have been performed with CAM-B3LYP[62]/def2-TZVP/CPCM[63]($CH_2Cl_2$) using SERENITY 1.5.2[64–66].

## Sample preparation for UV-Vis/Fluorescence/DLS measurements

All samples were prepared from a stock solution of (Z)**1** in $CHCl_3$ ($c = 2 \times 10^{-3}$ M) by transferring a defined amount into an empty glass vial, removing the solvent using an argon stream and dissolving the residue in the desired solvent to reach the desired concentration. The solution was subsequently transferred into a suitable glass cuvette and measured in the respective measuring device. Protonation and deprotonation experiments were carried out at $c = 1 \times 10^{-5}$ M with an initial volume of 3 mL and titrated against an acid/base solution ($c = 0.3$ M, 10 μL = 100 eq). Each addition step was accompanied by vigorous stirring (1000 rpm) for 30 s and recording of a full UV-Vis spectrum. Irradiation experiments were carried out in the respective UV cuvette by using an LED with a specified wavelength at a distance of 10 cm inside a closed box. For VT-UV-Vis and VT-luminescence experiments, the sample was initially heated to the initial target temperature (e.g., 380 K) and held at this temperature for 20 min to ensure a full molecularly dissolved state. Afterward, the sample was cooled down in a controlled manner (e.g., 1 K/min) while recording the UV-Vis spectra until the end temperature was reached.

## Sample preparation for solid-state NMR

The sample of (Z)**1** for the solid-state NMR experiments was prepared by dissolving a large amount of (Z)**1** (50 mg) in hot *n*-octane (20 mL, $1.3 \times 10^{-3}$ M) followed by slow cooling to room temperature. Subsequently, the solvent was removed by slow evaporation prior to the measurements. The sample of (Z)**1**-H⁺ for the solid-state NMR experiments was prepared by adding 1000 eq of TFA to a concentrated solution of (Z)**1**$_{Agg}$ in *n*-octane (55 mg in 20 mL) under slow stirring for 2 days. The solvent was removed by slow evaporation and the aggregate was obtained as an intense red solid.

## Data availability

The coordinates of optimized geometries are available in a separate Excel file as source data. All other data supporting the findings of this study, including experimental and synthetic procedures, compound characterization, theoretical calculations, UV-Vis, emission, NMR, DLS, MALDI-TOF, AFM and SEM analyses, are available within the article and its Supplementary Information or from the corresponding authors. Source data are provided with this paper.

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

## Acknowledgements

We thank the Deutsche Forschungsgemeinschaft (DFG, German Research Foundation)-Project-ID 433682494-SFB 1459 Intelligent Matter, Project A03) (T.D., N.N., S.B., S.H., M.R.H., J.N., G.F.) and the European Commission (ERC-StG-2016 SUPRACOP-715923) (T.D. and G.F.) for funding. We thank Timo Krüger and Dr. Zulema Fernández for fruitful discussions.

## Author contributions

T.D. and G.F. designed the project. T.D. performed the synthesis, UV-Vis, FT-IR, photoluminescence, DLS, NMR and SEM studies. S.B. performed supplementary UV-Vis and NMR studies. N.N. conducted all theoretical calculations. S.H. performed solid-state NMR studies. L.B. performed the AFM studies. T.D. and G.F. prepared the first draft of the manuscript including the figures, which was then revised and adapted upon contribution from all authors. The overall project was supervised by M.R.H., J.N. and G.F.

## Funding

## Competing interests

The authors declare no competing interests.
