## [Peer Review File · Nature Communications]

REVIEWER COMMENTS

Reviewer #1 (Remarks to the Author):

The authors fabricated a molecular system with a pH-responsive 2,2'-bipyridine and a photoresponsive cyanostilbene. The molecular structure could be regulated by the light or pH independently. In the aggregate state, the assembly morphology could be transformed between nanoparticles and 2D sheets with the introduction of acid or base. From a chemical perspective, numerous multi-stimuli-responsive systems have been fabricated by integrating different stimuli-responsive groups. The representative types of stimuli include force, light, temperature, and so on. In recent years, researchers are no longer satisfied with the construction of multi-stimuli-response systems, and the research focus has shifted towards endowing some functions (drug delivery, information encryption, bioimaging, etc.) to the system. In this work, it seems the system does not exhibit possible applications, and the stimuli-responsive groups have been widely used. What is more, I do not think this research provides new insights into "adaptation" from the molecular and supramolecular viewpoints. Based on this, I cannot recommend this manuscript for publication in Nature Communication. More comments are listed below:

- 1) In Figure 2b, the Z/E ratio is 69:31 at photostationary equilibrium and the arrow needs to be redrawn to reflect the ratio correctly.
- 2) The transformation and stability of the molecular in acidic and basic conditions should be investigated.
- 3) In Figure 3, different Z/E ratio could be obtained with different sequences of stimuli according to ¹H NMR spectra (Figure S23). It is not convincing for the difference between ¹H NMR spectra is not significance. The authors should explain the reason for the different Z/E ratio.
- 4) As the authors mentioned in the introduction "due to the lack of generally applicable definitions, the terms responsive and adaptive are oftentimes used interchangeably in the literature". What is the meaning of "adaptation" in this research? I do not think something new has been founded about the stimuli-responsive group utilized in this research.
- 5) It is hard for the readers to understand what the authors want to say. The paper should be rewritten to be more specific, especially the introduction.

Reviewer #2 (Remarks to the Author):

In this manuscript, Gustavo Fernández and coworkers have designed and synthesized compound (Z)-1 with a central pH-responsive 2,2'-bipyridine functionalized on either side with a photo-responsive cyanostilbene derivative bearing tris(dodecyloxy)benzene solubilizing groups. The judicious choice of two distinct stimuli-responsive moieties and molecular elements for

supramolecular self-assembly has allowed to elucidate the origin of adaptive behavior at both the molecular and supramolecular level. In particular, it undergoes molecular adaptation in good solvent as a result of coupled stimuli, leading to the Z/E isomer composition changes. In bad solvents, it displays reversible supramolecular adaptation upon the successive protonation/deprotonation, while irreversible supramolecular adaptation occurs to response to light. These very nice results shed light on the nontrivial concepts of coupled stimuli, conditioned responses and adaptation in chemical systems both from a molecular and supramolecular standpoint, which are key elements of intelligent matter. The conclusions are well backed up by the experimental evidences. Understanding these phenomena is thus a fundamental requirement for the development of advanced smart materials. My feeling is that this study contains important material that should be of interest to any potential reader of Nature Communications. Some minor issues should be addressed before the final acceptance of this manuscript (see below).

1. Fig. 2d: the pyridine resonances at 8.8 ppm underwent a down-field shift upon addition of 100 eq of TFA, and further displayed an up-field shift upon further addition to 1000 eq.. Similar phenomena exist for the pyridine resonances at 8.5 ppm. The authors are suggested to add a comment on the phenomenon, since it might suggest the involvement of two processes rather than direct protonation.

2. Figure 3: The electronic communication between the stimuli-responsive units is responsible to Z/E isomer composition changes. Is it possible to add some theoretical calculations to support the electronic communication effect?

3. Fig. 5b, inset: Cooling curves are employed to compare the thermodynamic parameters for (Z)1Agg and (Z)1-H+Agg. These parameters are suggested to be acquired by heating curves since heating curves are more prone to produce thermodynamic species than the cooling curves.

Reviewer #3 (Remarks to the Author):

This study demonstrates that different states can be obtained for a molecule that responds to both light and acid, depending on the order in which the stimuli are applied. The authors designed a molecule with cyanostilbene as the light-responsive moiety and a bipyridine moiety as the pH-responsive moiety. Spectroscopic and NMR studies have shown that the trans/cis ratio of the protonated molecule changes when the order of the two stimuli is changed, such as i) light and ii) acid or i) acid and ii) light. In addition, the authors have applied light and acid stimuli to molecules assembled in a nonpolar solvent. In the case of light, unlike the monomeric state, intermolecular photoreaction proceeded without photoisomerization, and a fiber-like network was obtained. In the case of acid, on the other hand, the self-assembled structure changed to sheet-like structures.

The responsive/adaptive behaviors, which depends on the aggregation state of the molecule, is interesting and has the potential to induce new physical properties by conducting similar

experiments on many molecules. In light of these results, this paper is suitable for publication in Nature Communications.

The following are suggestions from this reviewer to improve the manuscript and the scientific quality. If the authors have done any of the following experiments, the work would be enriched by adding some of the results.

(1) What would happen if the authors combined two stimuli for an assembled state that currently has only one stimulus applied?

(2) If the authors add nonpolar solvent to the monomeric state to which one of the stimuli has been already applied, what kind of aggregation state could be obtained? If the authors consider the addition of a low-polarity solvent as a stimulus, could they achieve a more complex molecular system? More specifically, this experiment can be performed also for the case where both stimuli have been already applied.

POINT-BY-POINT RESPONSE TO THE REVIEWERS' COMMENTS

Reviewer 1

In recent years, researchers are no longer satisfied with the construction of multi-stimuli-response systems, and the research focus has shifted towards endowing some functions (drug delivery, information encryption, bioimaging, etc.) to the system. In this work, it seems the system does not exhibit possible applications, and the stimuli-responsive groups have been widely used. What is more, I do not think this research provides new insights into adaptation from the molecular and supramolecular viewpoints. Based on this, I cannot recommend this manuscript for publication in Nature Communication.

We acknowledge the referee's feedback and would like to reply to his/her comments in the following way:

Our manuscript does not intend to develop applications, as it is a conceptual work where we define the notions of molecular and supramolecular responsiveness and adaptation. Kindly note that these non-trivial concepts have never been tackled as such in the literature, despite the fact that the terms responsive and adaptive are commonly (and often interchangeably) used since decades in the literature. Since 2020, and as part of a collaborative research center (CRC 1459) "intelligent matter", we and many other scientists from various disciplines ranging from mathematics, biology, physics, chemistry, and material sciences have been working intensively to rationalize key elements of intelligence, including the concepts of adaptation and responsiveness. The knowledge gained and the results achieved in the past four years by our teams (Profs. Neugebauer, Hansen and Fernandez alongside several coworkers) have been summarized in the current publication. In our opinion, we have found a general way to define responsiveness and adaptation in molecular and supramolecular systems, which, together with the technical quality of our work, represents a perfect match for Nature Communications. We appreciate the referee's feedback, which we have taken very seriously to revise the manuscript. In particular, we have expanded the discussion about adaptive and non-adaptive systems at the molecular level and highlighted these differences by including a new scheme in the manuscript (new Scheme 2) accompanied with a paragraph describing these differences in detail. We thank the referee for his/her feedback regarding this point, as now these concepts are not only clearer in the text, but also graphically illustrated.

1) In Figure 2b, the Z/E ration is 69:31 at photostationary equilibrium and the arrow needs to be redrawn to reflect the ratio correctly.

We thank the referee for highlighting this inconsistency, which we have addressed now in the revised manuscript.

2) The transformation and stability of the molecular in acidic and basic conditions should be investigated.

To test the stability of the molecule, we have performed time-dependent UV-Vis-experiments (1×10^{-5} M, CHCl_3) in the presence of either TFA or NEt_3 (1000 eq). Within the chosen timeframe of 24 h, no sign of side reactions, decomposition or change in the degree of protonation could be detected.

Additionally, the previously protonated state was subsequently deprotonated by addition of NEt_3 (1200 eq) and subsequently monitored for 24 h. As observed previously, no change could be observed. Thus, we conclude that the molecule is stable both in acidic and basic conditions. We have added the corresponding figures to the SI (Fig. S19) and inserted a short comment to the main text.

Fig. S19: a) Time-dependent UV-Vis spectra of **(Z)-1** (1×10^{-5} M, CHCl_3 , 298 K, 24 h) after addition of 1000 eq NEt_3 . b) Time-dependent UV-Vis spectra of **(Z)-1** (1×10^{-5} M, CHCl_3 , 298 K, 24 h) after addition of 1000 eq TFA. c) Time dependent UV-Vis spectra of **(Z)-1** (1×10^{-5} M, CHCl_3 , 298 K, 24 h) after the deprotonation of b) with 1200 eq NEt_3 .

3) In Figure 3, different Z/E ratio could be obtained with different sequences of stimuli according to ^1H NMR spectra (Figure S23). It is not convincing for the difference between ^1H NMR spectra is not significance. The authors should explain the reason for the different Z/E ratio.

We thank the reviewer for this comment. After carefully analyzing the NMR spectra, the differences in Z/E ratio, even if not very pronounced, are evident from the integration of the signals (80:20 vs. 69:31). The reason for the different Z/E ratio depending on the sequence of stimuli can be ascribed to the π -conjugation of the two stimuli responsive units, *i.e.* they are electronically coupled. As a result, protonation leads to a partial localization of the π -type HOMO and LUMO both in the Z- and in the E-isomer. As a result, differences in the absorption behavior close to the excitation wavelength are becoming less pronounced, *i.e.* the ratio $\epsilon(\text{Z})/\epsilon(\text{E})$ is reduced compared to the non-protonated case, in agreement with the absorption spectra in Figure 3 of the main text. As a consequence, the PSS shifts more towards the Z-form in the protonated case.

We have added a short discussion to the revised manuscript and included a figure showing the HOMOs and LUMOs of all relevant species as well as a table with calculated excitation energies and oscillator strengths as Fig. S25 and Tab. S2 in the revised supporting information.

4) As the authors mentioned in the introduction “due to the lack of generally applicable definitions, the terms responsive and adaptive are oftentimes used interchangeably in the literature”. What is the meaning of adaptation in this research? I do not think something new has been founded about the stimuli-responsive group utilized in this research.

We thank the referee for this comment, which has been previously addressed in our response to his/her first general comment. To address this point, we have clarified the differences between non-adaptive and adaptive molecular systems and included a new scheme to the revised main paper. Both comments from the referee have allowed us to improve the understanding and reinforce the main message of the manuscript.

5) It is hard for the readers to understand what the authors want to say. The paper should be rewritten to be more specific, especially the introduction.

We have adapted the introduction, added a new scheme and included more explanations, as discussed above in our replies to the referee's general concerns about our work and in our response to comment 4.

Reviewer 2

These very nice results shed light on the nontrivial concepts of coupled stimuli, conditioned responses and adaptation in chemical systems both from a molecular and supramolecular standpoint, which are key elements of intelligent matter. The conclusions are well backed up by the experimental evidences. Understanding these phenomena is thus a fundamental requirement for the development of advanced smart materials. My feeling is that this study contains important material that should be of interest to any potential reader of Nature Communications. Some minor issues should be addressed before the final acceptance of this manuscript (see below).

We thank the reviewer for the appreciation and positive evaluation of our work. His/her comments have been addressed as follows.

1) Fig. 2d: the pyridine resonances at 8.8 ppm underwent a down-field shift upon addition of 100 eq of TFA, and further displayed an up-field shift upon further addition to 1000 eq.. Similar phenomena exist for the pyridine resonances at 8.5 ppm. The authors are suggested to add a comment on the phenomenon, since it might suggest the involvement of two processes rather than direct protonation.

We thank the referee for raising this important issue. Based on the titration UV-vis experiments with TFA, we observe clear isosbestic points during the whole TFA addition, indicating the clean transition between two distinct species, *i.e.* direct protonation rather than the involvement of two processes. The observed phenomenon is most likely a consequence of the increased solvent polarity, due to the high amount of TFA- d_1 added in the NMR experiment, which has been previously observed in the literature (Chem. Eur. J. 2020, 26, 10005; Organometallics 2010, 29, 2176; Pharmaceutical Fronts 2023, 05, e288; Anal. Sci. 1995, 11, 631). Especially in larger π -conjugated systems with D-A-characteristics, the proton signals are sensitive to the polarity of the surrounding solvent. This effect can be seen in the shifting and separation of the signals of the aromatic core in THF- d_8 (used for the characterization) and in $CDCl_3$ (used for the protonation experiment). We have added a comment to the revised manuscript to address the referee's comment.

2) Figure 3: The electronic communication between the stimuli-responsive units is responsible to Z/E isomer composition changes. Is it possible to add some theoretical calculations to support the electronic communication effect?

As suggested by the referee, we have now performed theoretical calculations addressing the lowest excitation energies and the relevant orbitals involved in the corresponding electronic transitions for protonated and non-protonated species (see also answer to comment 3 of reviewer 1). We have

added a short discussion to the revised manuscript and included the results of these new calculations as Figures S25 and Tab. S2 in the revised supporting information.

3) Fig. 5b, inset: Cooling curves are employed to compare the thermodynamic parameters for $(Z)1_{\text{Agg}}$ and $(Z)1\text{-H}^+_{\text{Agg}}$. These parameters are suggested to be acquired by heating curves since heating curves are more prone to produce thermodynamic species than the cooling curves.

We thank the referee for this comment. In our initial submission, we demonstrated by cooling rate-dependent as well as by temperature-dependent UV/Vis studies that $(Z)1_{\text{Agg}}$ and $(Z)1\text{-H}^+_{\text{Agg}}$ are the only self-assembled species formed by $(Z)1$ and $(Z)1\text{-H}^+$, respectively, with no evidence of other competing kinetic species (Fig. S27, S28, Fig. 4b and Fig. 5c). As suggested by the referee, we repeated cooling experiments for both species $(Z)1$ and $(Z)1\text{-H}^+$, followed by a subsequent heating. In both cases, no sign of the formation of another competing aggregate could be observed, as evident from nearly identical spectral characteristics compared to cooling. We have added the additional experiments to the SI (Fig. S29 and S35) and included a short note in the main text.

Fig. S29: a) VT-UV-Vis-spectra of $(Z)1$ (1×10^{-4} M, 1 K/min, *n*-octane, 298 K) upon cooling. b) VT-UV-Vis-spectra of $(Z)1$ (1×10^{-4} M, 1 K/min, *n*-octane, 298 K) of the subsequent heating experiment.

Fig. S35: a) VT-UV-Vis-spectra of $(Z)1\text{-H}^+$ (1×10^{-5} M, 1 K/min, *n*-octane) upon cooling. b) VT-UV-Vis-spectra of $(Z)1\text{-H}^+$ (1×10^{-5} M, 1 K/min, *n*-octane) of the subsequent heating experiment.

We additionally recorded the respective heating curves, as also suggested by the reviewer. For the protonated species (**Z**1- H^+), the heating curves are nearly a superposition of the cooling ones, as shown in the images below, right. However, for the neutral form (**Z**1), heating experiments do not produce as well-defined curves as those obtained during cooling (see image below, left) and the fitting of these curves is not possible. It is evident from the identical spectral features upon heating and cooling (see above fig. S29), as well as by the cooling-rate independent curves previously included in the original submission that no kinetic species are formed, only the thermodynamic form. At this stage, the reasons for the lack of well-defined heating curves for (**Z**1), particularly at lower temperatures, are unclear. We speculate that one possible reason might be the changes in the conformation of the six long alkoxy chains upon heating, which are shielding the nanoparticle aggregates of (**Z**1) $_{\text{Agg}}$. Thus, given that heating curves cannot be used to obtain the thermodynamic data, we excluded them from the revised version of the manuscript and relied on the more accurate cooling curves. In fact, cooling curves are often used in the literature to calculate thermodynamic parameters (Chem. Eur. J. 2010, 16, 362; Chem. Rev. 2009, 109, 5687), as the slow cooling from the molecularly dissolved state enables the formation of ordered aggregates and favors thermodynamic equilibrium.

Fig.: Left: Heating curves of (**Z**1) in *n*-octane at different concentrations. Right: cooling and subsequent heating curve of (**Z**1- H^+) in *n*-octane at $1 \times 10^{-5} \text{ M}$ showing almost superimposable plots.

Reviewer 3

The responsive/adaptive behaviors, which depends on the aggregation state of the molecule, is interesting and has the potential to induce new physical properties by conducting similar experiments on many molecules. In light of these results, this paper is suitable for publication in Nature Communications.

We thank the reviewer for the appreciation and positive evaluation of our work. His/her comments have been addressed as follows.

The following are suggestions from this reviewer to improve the manuscript and the scientific quality. If the authors have done any of the following experiments, the work would be enriched by adding some of the results.

1) What would happen if the authors combined two stimuli for an assembled state that currently has only one stimulus applied?

We thank the reviewer for raising this important point. We had already applied the mutual stimuli to the respective assembled state in our initial submission, but unfortunately, this discussion was not sufficiently emphasized in the main text. By addition of acid or base to the obtained polymer network that was obtained after irradiation of **(Z)1**_{Agg}, no further reaction could be observed, as the polymer precipitate (**(Z)1**_{Photo}) is no longer soluble. Thus, once **(Z)1**_{Agg} has been exposed to light, the system remains inert and cannot respond to subsequent stimuli, such as TFA. On the other hand, if the protonated aggregate **(Z)1-H**⁺_{Agg} is subjected to irradiation, a decomposition reaction was detected (see Fig. S43). We have now highlighted these results more clearly in the revised manuscript.

2) If the authors add nonpolar solvent to the monomeric state to which one of the stimuli has been already applied, what kind of aggregation state could be obtained? If the authors consider the addition of a low-polarity solvent as a stimulus, could they achieve a more complex molecular system? More specifically, this experiment can be performed also for the case where both stimuli have been already applied.

This is a very helpful and important suggestion, which is in fact currently under investigation in our group. As correctly mentioned by the reviewer, inducing aggregation by a poor solvent after application of stimuli is a powerful method to create more complex energy landscapes due to the formation of multiple kinetic species. In fact, given that our system **(Z)1** can exist in different molecular states (protonated, neutral, *Z* and *E*) and at different ratios, the energy landscape becomes very complex. We are currently analyzing the combined effect of using low-polarity solvents as stimuli, together with temperature, acid, base and light irradiation to create advanced supramolecular adaptive systems for the current molecule **(Z)1** as well as other derivatives bearing pyridines instead of bipyridines. We plan to combine all these studies at the supramolecular level and submit them to publication elsewhere in the near future, as the focus will be exclusively placed on supramolecular adaptation.

To address the referee's comment, we have performed the suggested experiments of inducing aggregation by a low-polarity solvent upon application of the single stimuli. In general, we firstly applied the respective stimulus (irradiation with $\lambda_{LED} = 465$ nm or TFA addition) in CHCl_3 and enforced aggregation by addition of *n*-octane. First, we monitored this effect upon irradiation. To this end, we dissolved **(Z)1** in CHCl_3 (1×10^{-4} M), irradiated with $\lambda_{LED} = 465$ nm, removed the solvent under an argon stream, added the same amount of *n*-octane (3 mL) to reach a concentration of 1×10^{-4} M and performed a cooling experiment. In this way, the mixture of isomers (*Z/E* 69:31) is subjected to aggregation, which is monitored by UV-vis spectroscopy. The aggregation is characterized by a decrease in λ_{max} and the rise of the absorbance at lower energies (see below). By comparing the aggregation curves of **(Z)1** and the 69:31 **(E)1:(Z)1** mixture, we observed a decrease in the elongation temperature of 12 K for the mixture compared to the pure **(Z)1** form. We attribute the attenuated aggregation tendency of the mixture to increased steric effects due to the presence of the distorted *E*-isomer units, in accordance with the literature (Chem. Sci., 2019,10, 752). As mentioned, we are currently examining in detail the combined use of nonpolar solvents as chemical stimuli and their combined effect together with with TFA and light for **(Z)1**, along with other bipyridine and pyridine-

based chromophores. As these studies are already ongoing and require considerable experimental work to understand the intricacies of these systems, this topic will be published separately and is out of the scope of this manuscript. Exemplarily, the reviewer can find below first investigations of the comparison between **(Z)1** and the 69:31 **(E)1**:**(Z)1** mixture, which, for the reasons explained above, will not be included in the present manuscript.

Fig.: a) UV-Vis spectra of **(Z)1** (1×10^{-5} M, CHCl_3 , 298 K) under irradiation with $\lambda_{\text{LED}} = 465$ nm. Inset: Plot of the ϵ at λ_{max} vs. the irradiation time. b) UV-Vis cooling experiment of this 69:31 **(E)1**:**(Z)1** mixture in *n*-octane, showing differences compared to pure **(Z)1**_{Agg}. c) comparison of the cooling curves for the aggregates of pure **(Z)1** (black plot) and the corresponding 69:31 **(E)1**:**(Z)1** mixture (red plot).

In a second approach, we carried out a solvophobic quenching experiment (Chem. Sci. 2020, 11, 6780), in which 30 μL of **(Z)1** in CHCl_3 (1×10^{-3} M) were protonated with TFA (100 μL , 0.3 M, 1000 eq) and the resulting protonated **(Z)1-H⁺** (in the molecularly dissolved state) was added into a large volume of *n*-octane (3 mL). The resulting UV-Vis spectrum clearly resembles the characteristic features of the already investigated **(Z)1-H⁺**_{Agg}, indicating the absence of another aggregated species (Fig. S36).

Fig. S36: a) UV-Vis spectra of **(Z)1-H⁺**_{Agg} (1×10^{-5} M, *n*-octane, 298 K) obtained by various techniques (titration, solvophobic quenching, cooling). b) Normalized UV-Vis-spectra of **(Z)1-H⁺**_{Agg} (1×10^{-5} M, *n*-octane, 298 K) obtained by various techniques (titration, solvophobic quenching, cooling).

To follow the transition of $(Z)1-H^+_{Agg}$ to $(Z)1-H^+$ in different solvent ratios of $CHCl_3$ and n -octane, we also performed a denaturation experiment (J. Am. Chem. Soc. 2012, 134, 13482) at $c = 1 \times 10^{-5}$ M. With increasing volume percentages of good solvent ($CHCl_3$), a clean transition to the disassembled species $(Z)1-H^+$ can be observed (Figure S36). Furthermore, by adding 100 eq of NEt_3 , the neutral $(Z)1$ can be retained. The overall experimental results suggest that the formation of the protonated aggregate $(Z)1-H^+_{Agg}$ is independent of the sample preparation method, *i.e.* whether the acid is added in the aggregated or molecularly dissolved state. On the other hand, irradiation of $(Z)1$ in a monomeric state and subsequent induction of aggregation by addition of a poor solvent leads to a different outcome compared to the irradiation of the assembly $(Z)1_{Agg}$. The according UV-Vis spectra have been added to the SI (Fig. S36, S37) and a short discussion has been added to the main text.

Fig. S37: a) UV-Vis spectra of the denaturation experiment of $(Z)1-H^+_{Agg}$ (1×10^{-5} M, n -octane, 298 K). The experiment was conducted by the addition of incremental amounts of an equimolar monomeric solution of $(Z)1-H^+_{Agg}$ in $CHCl_3$ to the aggregate solution to achieve concentration consistency. Inset: Plot of α_{Agg} vs. volume fraction of $CHCl_3$. b) UV-Vis spectra of the consecutive deprotonation with NEt_3 .

REVIEWERS' COMMENTS

Reviewer #1 (Remarks to the Author):

The authors have made significant improvements. The molecule is stable in acidic or basic conditions indicating the absorption transformation is attributed to the change of pH. A possible mechanism is also proposed to explain the relationship between the final state and the stimuli sequences. In addition, the authors give an expanded discussion about adaptive and non-adaptive systems at the molecular level as shown in Scheme 2, and it is helpful for the readers to understand the difference between adaptation and responsiveness. While the difference between the final state is not significant (80:20 and 69:31), more high efficiency systems need to be designed and fabricated to interpret the concept of adaptation. Therefore, I would like to recommend this manuscript for publication in Nature Communication after revisions as follows.

- 1) The proposed mechanism should be discussed with more details in the manuscript.
- 2) Does the wavelength of light influence the final state?
- 3) Based on the mechanism proposed in this study, how to enlarge the difference between the final states?

Reviewer #2 (Remarks to the Author):

The authors have addressed all of the issues raised by the reviewers. The quality of the revised manuscript has been considerably improved.

Moreover, this research shed light on the nontrivial concepts of coupled stimuli, conditioned responses and adaptation in chemical systems both from a molecular and supramolecular standpoint, representing key elements of intelligent matter, which have never been tackled in the previous literatures. Given the novel concept together with the technical quality of the work, I strongly recommend the acceptance of the manuscript in the current form in Nature Communications.

Reviewer #3 (Remarks to the Author):

The authors kindly responded to my comments, which were made out of genuine interest, and I am grateful for that. I look forward to seeing these results develop further. I believe this paper should be published in Nature Communications.

POINT-BY-POINT RESPONSE TO THE FINAL COMMENTS RAISED BY REVIEWER 1

1) The proposed mechanism should be discussed with more details in the manuscript.

According to this comment, we have expanded the discussion about the differences in Z/E ratio depending on the sequence of stimuli. A new paragraph has been added to the main paper.

2) Does the wavelength of light influence the final state?

We thank the referee for this comment. Kindly note that these experiments were included in our previous submission, Supplementary images 20 and 21 (now supplementary images 30 and 31), showing for all wavelengths a higher population of the Z-isomer upon protonation compared to the neutral form. We have added a new sentence to the final manuscript to highlight these results. Please also note that we are rather restricted in terms of irradiation wavelengths and times (as also discussed in the text), as even wavelengths of 430 nm can quickly lead to other side photoreactions, especially after longer irradiation times. This is the reason why we mostly focused on 465 nm.

3) Based on the mechanism proposed in this study, how to enlarge the difference between the final states?

This is a very important comment. Based on ongoing studies and many other molecules that are currently under investigation, we can conclude that the efficiency of the photoisomerization (which is directly related to the differences between the final states) can be improved by reducing the size of the π -system and using one cyanostilbene unit instead of two. For some of these dual acid and light-responsive molecules, we have found efficiencies for the Z-to-E-photoisomerization of more than 90%. We have added a new sentence at the end of the manuscript in the discussion section to highlight how the photoisomerization efficiency can be improved. These results will be submitted to publication soon as a separate manuscript.